# Efficacy of Carbon Nanodots and Manganese Ferrite (MnFe_2_O_4_) Nanoparticles in Stimulating Growth and Antioxidant Activity in Drought-Stressed Maize Inbred Lines

**DOI:** 10.3390/plants12162922

**Published:** 2023-08-11

**Authors:** Muhammad Zahaib Ilyas, Hyeon Park, Young Sun Baek, Kyu Jin Sa, Myong Jo Kim, Ju Kyong Lee

**Affiliations:** 1Department of Applied Plant Sciences, College of Agriculture and Life Sciences, Kangwon National University, Chuncheon 24341, Gangwon-do, Republic of Korea; zahaib1996@kangwon.ac.kr (M.Z.I.); hyeonpark@kangwon.ac.kr (H.P.); sakajin49@kangwon.ac.kr (K.J.S.); kimmjo@kangwon.ac.kr (M.J.K.); 2Interdisciplinary Program in Smart Agriculture, Kangwon National University, Chuncheon 24341, Gangwon-do, Republic of Korea; sunnybaek96@kangwon.ac.kr

**Keywords:** drought stress, maize seedling, carbon nanodots, MnFe_2_O_4_, HPLC, phenolic compounds

## Abstract

Despite being the third most-consumed crop, maize (*Zea mays* L.) is highly vulnerable to drought stress. The predominant secondary metabolite in plants is phenolic acids, which scavenge reactive oxygen species to minimize oxidative stress under drought stress. Herein, the effect of carbon nanodots (CND) and manganese ferrite (MnFe_2_O_4_) nanoparticles (NP) on the drought stress tolerance of maize has been studied. The experimental results revealed that the highest leaf blade length (54.0 cm) and width (3.9 cm), root length (45.2 cm), stem diameter (11.1 mm), root fresh weight (7.0 g), leaf relative water content (84.8%) and chlorogenic (8.7 µg/mL), caffeic (3.0 µg/mL) and syringic acid (1.0 µg/mL) contents were demonstrated by CND-treated (10 mg L^−1^) inbred lines (GP5, HW19, HCW2, 17YS6032, HCW3, HCW4, HW7, HCW2, and 16S8068-9, respectively). However, the highest shoot length (71.5 cm), leaf moisture content (83.9%), shoot fresh weight (12.5 g), chlorophyll content (47.3), and DPPH free radical scavenging activity (34.1%) were observed in MnFe_2_O_4_ NP-treated (300 mg L^−1^) HF12, HW15, 11BS8016-7, HW15, HW12, and KW7 lines, respectively. The results indicate that CND and MnFe_2_O_4_ NP can mitigate drought stress effects on different accessions of the given population, as corroborated by improvements in growth and physio-biochemical traits among several inbred lines of maize.

## 1. Introduction

Water is an inevitable input to agriculture. However, with climate changes and increasing groundwater limitations for crop irrigation, drought stress has emerged as a serious threat to crop production [1,2]. It is predicted that 30% of global water resources will be diminished, and drought-prone regions will double by 2050 [3]. Conversely, market requirements for agricultural produce, including cereals, are projected to grow by 50% by 2030 [4].

After rice (*Oryza sativa* L.) and wheat (*Triticum aestivum* L.), maize (*Zea mays* L.) is ranked as the third most consumed crop in the world, contributing both to food security and economic development [4,5]. It is a crop that is highly sensitive to drought stress, especially at critical growth stages such as the seedling stage [6,7,8]. Therefore, it is often used as an ideal crop to assess drought tolerance [9]. Moreover, there is substantial documentation that maize appears to be more responsive to drought stress than other cereal crops [10]. 

When a plant undergoes drought stress, the key indicator is the lowered turgor pressure of aerial plant parts. This leads to reduced cell division and elongation [11]. Drought stress substantially affects plant growth, development, and agronomic traits by disrupting physiology and anatomical structure. It disturbs source-sink relationships, stomatal gaseous exchange, plant-water relations, nutrient transport and assimilation, osmotic balance, and several metabolic pathways in plants [12]. Drought stress dramatically reduces photosynthetic activity by decreasing CO_2_ diffusion from the environment to the carboxylation site and leaf chlorophyll content [13,14].

The immediate effect of water deficiency stress is a disequilibrium between the production of reactive oxygen species (ROS) and their scavenging [15]. Drought-induced ROS generation occurs in various cell compartments, including chloroplasts, peroxisomes, and cell membranes [16]. ROS production in plants occurs through the reduction of oxygen (O_2_) into superoxide (O_2_^•−^), hydrogen peroxide (H_2_O_2_), hydroxyl radical (HO^•^), and singlet oxygen (^1^O_2_) [17]. In addition to damaging the proteins, nucleic acid, and lipids of cells, high levels of these ROS adversely affect stomatal activity, signal transduction, the electron transport chain, and the seed set [18,19,20].

Nevertheless, plants also have a well-developed antioxidant defense system to minimize the oxidative stress caused by excessive ROS production [21]. This antioxidant system is comprised of enzymatic, i.e., ascorbate peroxidase (APX), peroxidase (POD), superoxide dismutase (SOD), catalase, etc., and non-enzymatic, i.e., ascorbic acid (AsA), phenolic compounds, glutathione (GSH), oxidized glutathione (GSSG), etc., antioxidants, which together may help plants to cope with drought stress [21,22]. 

Non-enzymatic antioxidants, particularly phenolic compounds, are considered critical defense compounds under stressful environmental conditions [23]. Phenolics are secondary metabolites (esters, flavonoids, hydroxycinnamate, lignin, and tannins) found in different plant tissues [24], and their high accumulation is considered a distinct plant stress trait [23]. This accumulation results from the activity of chalcone synthase (CHS), phenylalanine ammonia-lyase (PAL), and other related enzymes [23]. According to Robards and Antolovich [25], around 2% of all carbon photosynthesized by plants is transformed into phenolics. Plant phenolic compounds are biosynthesized using a biosynthetic intermediate, i.e., shikimic acid and phenylalanine, via the shikimic acid pathway [23]. 

To minimize drought-induced oxidative damage and enhance the antioxidant potential of plants, treatment by nanoparticles (NP) is one of the most effective techniques [26]. Recent studies have reported that nanoparticles directly influence plant physiological events. This promotes plant growth, development, and tolerance by inducing seed germination, upregulating the antioxidant system, promoting nutrient absorption, improving photosynthesis, and boosting overall crop productivity [27,28,29]. 

Recently, agricultural applications of carbon nanomaterials have gained attention because of their unique structural and physical properties. Several forms of carbon nanomaterials are available; however, carbon nanodots (CND) have shown remarkable promise for improving growth and yield by augmenting the photosynthetic efficiency of both C4 and C3 plants [30]. When 5 mg L^−1^ carbon dots were sprayed on maize leaves, their fluorescence was observed around the chloroplast using a confocal laser scanning microscope (CLSM), which confirmed their uptake and translocation from the leaf surface [31]. CND can also enhance ROS scavenging by boosting the antioxidant system, root activity, chlorophyll content, and biomass accumulation, resulting in improved plant resistance to abiotic stress. Moreover, their slightly acidic nature and electronegative functional groups can offer a negative charge to the surrounding medium, activating biomacromolecule functions and nutrient ions, while their nanoscale structure and ample hydrophilic functional groups facilitate nutrient and water delivery to plant organs, accelerating plant growth [32,33]. 

Spinel ferrites are another type of nanomaterial composed of metal oxides with spinel structures, and their general chemical formula is AB_2_O_4_, where “A” represents a divalent cation (Mn^2+^, Ni^2+^, Zn^2+^, Fe^2+^) and “B” represents a trivalent cation (Fe^3+^, Mn^3+^) [34,35]. Manganese ferrite (MnFe_2_O_4_) is a well-known spinel ferrite that exhibits strong chemical stability, soft magnetic properties, and simple preparation [36]. According to their dimensions, MnFe_2_O_4_ NP exhibit lower magnetization and higher coercivity than bulk MnFe_2_O_4_ [37]. MnFe_2_O_4_ NP have been used effectively as a contrast agent in magnetic resonance imaging (MRI), mediators in cancer thermotherapy [38], and to remove heavy metals, polychlorinated biphenyls, chlorinated organic compounds, and numerous other inorganic and organic compounds from contaminated water and soil [39]. 

Manganese (Mn) and iron (Fe) are crucial for plant photosynthesis [40]. So, MnFe_2_O_4_ NP were selected with the anticipation that the foliar application of such composite nanomaterials, which contain both Mn and Fe, can enhance plant growth and alleviate drought stress. It was typically found that MnFe_2_O_4_ NP are between 20 and 60 nm in size [41]; however, the size of plant stomata ranges from 10 to 100 µm, so leaves are capable of fully absorbing these nanoparticles [42]. A study conducted on barley found that MnFe_2_O_4_ NP treatment improved seed germination, plant growth, and biomass, with the highest growth rate at 250 mg L^−1^ of MnFe_2_O_4_ NP application. However, higher doses of MnFe_2_O_4_ NP hindered barley growth [43]. In another study, MnFe_2_O_4_ NP were applied foliarly to tomato plants to investigate their effects on the vegetative and reproductive stages. The results indicated that MnFe_2_O_4_ NP appear to act as an electron donor to promote photosynthetic electron transport, early flowering induction, enhanced pollen activity, ovule size and fruit weight in tomato [41]. MnFe_2_O_4_ NP also increased the nutritional value of tomato fruits by increasing glucose-6-phosphate, rutin, phenylalanine, and vitamin C and reducing methionine and tomatine levels [41].

To our knowledge, there have been no studies reporting abiotic (including drought) stress mitigation in maize through MnFe_2_O_4_ NP, whereas only a few studies have investigated drought stress alleviation in maize by CND [31,32,33,34,35,36,37,38,39,40,41,42,43,44]. However, these studies focused mainly on the effect of CND on photosynthesis and carbon metabolism without exploring plant metabolites such as phenolics. In this study, we hypothesize that foliar application of CND and MnFe_2_O_4_ NP to drought-stressed maize inbred lines will enhance their drought tolerance by increasing their morphological, physiological, and biochemical responses. To the best of our knowledge, no information is available on phenolic acids (hydroxycinnamic acids (HCAs) and hydroxybenzoic acids (HBAs)) in CND or MnFe_2_O_4_ NP-treated drought-stressed cereals. So, the findings of this study will serve as the basis for future research involving secondary metabolites as well as provide insight into the potential of these nano chemicals to alleviate drought stress in the maize population.

## 2. Results

### 2.1. Seed Germination

Germination percentages of the 41 elite maize inbred lines employed in the current study (Figure 1) show that, out of the 41 inbred lines, 13 resulted in 100% germination, namely 14S8025, 16S8068-9, 17CS8006, 17CS8067, 17YS6032, 17YS8003, GP3, GP5, HF22, HW12, HW16, HW3, and HW9. There were eight inbred lines that demonstrated the same germination percentage (94.4%), namely 15RS8056, 15S8021-3, HCW2, HCW5, HW1, HW17, HW4, and KL103. Among the 41 lines, 12BS5076-8 and KW7 showed the lowest germination rates (27.7%). To verify the results of these two lines, another experiment was run under controlled germination conditions, and the same results were obtained. The germination percentage of the rest of the inbred lines used in this study ranged from 55 to 88%.

### 2.2. Effects of CND and MnFe_2_O_4_ NP on Plant Growth and Morphology under Drought Stress

#### 2.2.1. Leaf Blade Length and Width

The effect of CND and MnFe_2_O_4_ NP on the leaf blade length of the 41 inbred lines under drought stress was investigated (Table 1). The GP5 inbred line exhibited the longest leaf blade of 54 cm when treated with CND (10 mg L^−1^) under drought stress, which is 23.2% longer than the control. Two inbred lines, 15RS8056 and 17CS8067, displayed the highest compatibility with the CND application and showed a statistically significant increase in leaf blade length of 36.3 and 38.8%, respectively, over the control conditions. Moreover, line KL103 showed the highest increase of 28.5% in leaf blade length among the 41 lines treated with MnFe_2_O_4_ NP compared with the controls. In contrast, 11 lines (16S8068-9, 17CS8006, 17YS8003, HF12, HW1, HW11, HW17, HW18, HW4, HW7, and KW7) were negatively affected by CND and MnFe_2_O_4_ NP application under drought stress with their leaf blade length decreasing from −0.5 to −23.0% compared with the controls. 

Results of the impact of MnFe_2_O_4_ NP and CND on the leaf blade width of the 41 inbred lines under drought stress (Table 1) showed that the drought-stressed HW19 inbred line had the highest leaf width (3.9 cm) under CND treatment, with a statistically significant increase of 25.8% compared with the control. In the HW3 line, CND and MnFe_2_O_4_ NP had the highest synergistic effect on leaf width, with statistically significant increases of 102.1 and 135.4%, respectively, over the control. Conversely, some inbred lines also responded negatively (14S8025, 17CS5047, 17CS8067, 17YS8003, HW17, etc.) for leaf blade width to foliar applications of CND and MnFe_2_O_4_ NP, with decreases ranging between −1.2 and −32.6% compared with their respective controls. 

#### 2.2.2. Shoot and Root Length

Lines HF12 and 12BS5076-8 had the longest shoot (71.5 cm) and the highest compatibility (50.7% increase in shoot length), respectively, under MnFe_2_O_4_ NP application treatment (Table 2). Meanwhile, line 15RS8056 exhibited the highest increase in shoot length (35.0%) as a result of CND treatment. The following inbred lines responded negatively to both CND and MnFe_2_O_4_ NP applications: 16CLP23, 17CS5047, 17YS8003, HW11, HW8, and KW7.

The longest root (45.2 cm) was recorded in the HCW2 line when treated with CND (Table 2). Under MnFe_2_O_4_ NP treatment, the maximum root length (37.0 cm) was found in the 16CLP40 line. When compared with the control, the greatest and statistically significant improvement in root length (177.2%) was observed in the 12BS5076-8 line treated with MnFe_2_O_4_ NP, while the second highest increase in root length was recorded in the HW9 line (102.7%) treated with CND. The application of both CND and MnFe_2_O_4_ NP resulted in antagonistic effects on root length in lines 15RS8002, 17YS6032, GP3, HF22, HW16, HW19, and HW7.

#### 2.2.3. Leaf Water Status and Stem Diameter

The leaf moisture contents of 41 inbred lines differed markedly but were statistically non-significant (Figure 2a). Under drought stress, the highest leaf moisture content (84.0%) was recorded in the HW15 line when treated with MnFe_2_O_4_ NP (Figure 2a). Among the 41 inbred lines, HW15 and 15RS8056 treated with MnFe_2_O_4_ NP showed the highest (73.9%) and second highest (31.3%) increases, respectively, in leaf moisture content under drought stress, compared with the control. Conversely, four inbred lines (14S8025, 15RS8039, HCW2 and HCW5) were negatively affected by foliar application of MnFe_2_O_4_ NP and CND, and their leaf moisture content was reduced.

The HCW4 line demonstrated the highest leaf relative water content (84.8%) when treated with CND, followed by the 15RS8056 line (84.0%) when treated with MnFe_2_O_4_ NP, and both were statistically significant when compared to controls (Figure 2b). Among the 41 inbred lines, HW15 treated with MnFe_2_O_4_ NP showed the highest increase in leaf relative water content (57.4%) compared with the control. Conversely, seven inbred lines (14S8025, 15RS8039, 16CLP23, HCW5, HW1, HW3 and KL103) were negatively affected by foliar application of CND and MnFe_2_O_4_ NP, and their leaf relative water content was reduced.

The highest leaf water saturation deficit was observed in HW15 (46.9%) from the control group, followed by the 14S8025 line (42.9%) when treated with CND, both of which were statistically significant (Figure 2c). Conversely, two MnFe_2_O_4_ NP-treated inbred lines (16CLP23 and KL103) and five CND-treated lines (14S8025, 15RS8039, HCW5, HW1, and HW3) were negatively affected by their foliar treatments, resulting in an increased leaf water saturation deficit.

Line 17YS6032 exhibited the highest stem diameter value (11.1 mm), and line 15RS8056 showed the maximum promotion (84.9%) under CND treatment (Figure 3), and both were statistically significant when compared to the controls. Moreover, line HW12 displayed both a maximum stem diameter of 10.3 mm and a promotion of 43.0% under the MnFe_2_O_4_ NP application. Along with the positively responding lines, a few lines that were negatively affected by CND and MnFe_2_O_4_ NP applications under drought stress were also found (Figure 3).

#### 2.2.4. Root and Shoot Fresh Weight

The line HCW3 exhibited the highest root fresh weight (7.0 g) when treated with CND, which was statistically significant compared to the control under drought stress (Table 3). The maximum increase in root fresh weight (275.5%) was observed in line 15RS8056 under CND treatment. More than half of the lines employed in this study showed a decrease in root fresh weight (ranging between −1.9 and −47.9%) upon MnFe_2_O_4_ NP application compared with their respective controls. Overall, in this experiment, CND was observed to improve root growth more than MnFe_2_O_4_ NP under drought-stress conditions.

The MnFe_2_O_4_ NP-treated HW15 line showed the maximum shoot-fresh weight (12.5 g) under drought stress, while line KL103 showed the highest increase (73.8%) in shoot-fresh weight under MnFe_2_O_4_ NP treatment; both were statistically significant compared to the controls (Table 3). The maximum shoot fresh weight (11.3 g) and highest increase (64.4%) under CND treatment were observed in lines 15RS8002 and KW7, respectively. In general, among the 41 lines, there was a better performance for shooting fresh weight under MnFe_2_O_4_ NP treatment compared with the CND application; however, a few lines showed negative responses to the CND and MnFe_2_O_4_ NP sprays.

### 2.3. Chlorophyll Content

In comparison with the controls, the HW12 line treated with MnFe_2_O_4_ NP had the highest leaf chlorophyll content (47.3), followed by the HCW1 line (45.5) treated with CND; both were statistically significant (Figure 4). Moreover, the same MnFe_2_O_4_ NP-treated HW12 line showed the greatest improvement of 64.6% in chlorophyll content, followed by the CND-treated HCW1 line with an improvement of 39.8%. Out of the 41 lines treated with MnFe_2_O_4_ NP, only six lines showed a decline in chlorophyll content, which ranged from −0.44 to −14.1%. Meanwhile, 17 lines exhibited a minimal reduction in chlorophyll content under CND treatment ranging between −1.1 and −14.2%. In summary, MnFe_2_O_4_ NP proved more effective for chlorophyll content enhancement under drought stress than CND treatment.

### 2.4. DPPH Free Radical Scavenging Activity

The drought-stressed 41 inbred lines showed a highly variable DPPH free radical scavenging potential (Figure 5). Two of the highest values for DPPH free radical scavenging potential were recorded in the MnFe_2_O_4_ NP-treated KW7 (34.1%) and 17YS6032 (27.5%) lines. A maximum improvement in scavenging potential was measured under the same MnFe_2_O_4_ NP treatment for the HW16 line (2373.4%), followed by the 17CS8006 line (2281.6%). Under CND treatment, line HW16 had the highest scavenging potential (1542.5%) for DPPH free radicals, followed by line HW17 (1465.3%). In the control group, 18 lines did not show the potential to scavenge DPPH free radicals, which was reduced to five lines under CND treatment and eight lines upon MnFe_2_O_4_ NP application. 

### 2.5. Total Phenolic Contents (TPC)

TPC of the 41 drought-stressed elite maize inbred lines was expressed as mg GAE/g sample (Figure 6). Surprisingly, drought-stressed line 12S8052 without nanoparticle application (control) exhibited the highest TPC (179.2 mg GAE/g), followed by the same line 12S8052 (170.7 mg GAE/g) under MnFe_2_O_4_ NP application. The third highest TPC (169.6 mg GAE/g) was found in the CND-treated GP5 line, which was statistically significant compared to the control. Meanwhile, the greatest improvement in TPC compared with the control was recorded in the 17CS8067 line (115.2%) under MnFe_2_O_4_ NP treatment, followed by the CND-treated HW9 line (104.7%); both were statistically significant. Among the 41 lines, 15 showed a decline in TPC when treated with MnFe_2_O_4_ NP under drought stress, with the decline ranging from −0.8 to −75.8%; while, under CND treatment, 16 lines showed declines ranging from −0.6 to −71.2%. 

### 2.6. HPLC-UV Analysis of Phenolic Compounds

Concentrations of six phenolic acids, viz., gallic acid, chlorogenic acid, caffeic acid, syringic acid, *p*-coumaric acid, and ferulic acid, were determined by the HPLC-UV analysis (Figure 7). Gallic acid showed totally different results from those of all the other traits presented so far. The highest concentration of gallic acid was measured in line 17CS8006 (5.4 µg/mL) in the control group, followed by line HWI (5.1 µg/mL), also in the control group; both were statistically significant (Appendix A). Moreover, only two lines (12BS5076-8 and GP3) exhibited an increase in gallic acid content when treated with MnFe_2_O_4_ NP, whereas 12 lines showed an increase under CND treatment. Overall, the control group performed better for gallic acid accumulation under drought stress than foliar treatment with CND or MnFe_2_O_4_ NP. 

In the case of chlorogenic acid, line HW7 showed no accumulation under both control and MnFe_2_O_4_ NP; however, when this line was treated with CND, it showed the maximum concentration of chlorogenic acid (8.7 µg/mL) among the 41 lines employed under the three treatments, and this is considered the best result. The greatest improvement in chlorogenic acid accumulation was observed in the CND-treated HF12 line (638.9%) under drought stress, which was statistically significant (Appendix A). The application of CND and MnFe_2_O_4_ NP negatively affected chlorogenic acid accumulation in a total of 13 and 29 maize inbred lines, respectively. In summary, CND treatment proved efficient in greatly enhancing chlorogenic acid accumulation in drought-stressed maize inbred lines.

For caffeic acid, the highest level was observed in the CND-treated HCW2 line (3.0 µg/mL), followed by the 16CLP23 line (2.9 µg/mL) from the control group. A maximum increase in caffeic acid accumulation under drought conditions was recorded in line HCW1 (320.7%) following CND treatment, which was statistically significant (Appendix A). When treated with MnFe_2_O_4_ NP, a total of 26 lines showed a decrease in caffeic acid content, whereas only 10 lines showed a decrease when sprayed with CND. In conclusion, the CND application to drought-stressed maize lines resulted in a notable increase in caffeic acid accumulation.

For syringic acid, the highest level was observed in the CND-treated 16S8068-9 line (1.0 µg/mL), followed by the GP5 line (1.0 µg/mL) with the same treatment; both were statistically significant compared to their corresponding controls (Appendix A). The maximum and statistically significant increase of 86.8% was also recorded in line 16S8068-9 following CND treatment under drought-stress conditions. There were 21 lines that showed a reduction in syringic acid accumulation after MnFe_2_O_4_ NP treatment and 18 lines that showed a reduction after CND treatment.

It has been observed that *p*-coumaric acid accumulation in maize lines under drought stress resembles that of gallic acid in some respects. Two of the highest *p*-coumaric accumulation levels were observed in the KW7 (4.1 µg/mL) and KL103 (3.7 µg/mL) inbred lines of the control group. Only one line (12BS5076-8) showed improvement in *p*-coumaric content upon MnFe_2_O_4_ NP spray, whereas 27 lines showed downregulation in *p*-coumaric content under CND treatment. In summary, the 41 lines tested in this study showed better accumulation of *p*-coumaric acid under the control than with the nanoparticle treatments. 

The most significant improvement in ferulic acid accumulation under drought stress was observed in line 17CS8067 (121.7%) following CND treatment, which was statistically significant (Appendix A). However, two of the highest levels of ferulic acid were found in the control group for lines HCW2 (1.8 µg/mL) and HW7 (1.8 µg/mL). Four lines of the control group exhibited no ferulic acid accumulation, namely, 12BS5076-8, 15RS8002, HW18, and HW9. However, when these lines underwent MnFe_2_O_4_ NP application, the HPLC analysis showed that all four lines produced ferulic acid. 

## 3. Discussion

In the current study, CND and MnFe_2_O_4_ NP were evaluated for their effectiveness in alleviating drought stress in 41 elite maize inbred lines by assessing their effects on morphological, biochemical, and physiological parameters. Germination is a crucial phase of a plant’s life cycle, especially for annual species subject to competitive conditions [45,46]. Developing novel varieties and hybrids requires the screening and inclusion of genotypes with high germination percentages [47]. As mentioned in the Results Section, there were high germination percentages for the majority of the 41 maize lines (Figure 1), which is consistent with another study conducted on 16 rice varieties under normal conditions [48]. The endogenous plant hormones gibberellic acid (GA) and abscisic acid (ABA) are the primary factors that regulate seed dormancy and germination. In particular, low GA/ABA concentrations may trigger seed dormancy, leading to low germination [48]. Moreover, environmental factors such as light, temperature, soil moisture and pH are known to influence seed germination [49]. 

The scarcity of available information regarding the effects of CND and MnFe_2_O_4_ NP on the morphological, physiological, and biochemical attributes of drought-stressed maize led us to compare our study with other related studies. Itroutwar et al. [50] reported that the highest leaf width (16 mm) and length (60 mm) were recorded in maize plants treated with 100 mg/L biogenic ZnO nanoparticles. This supports our findings demonstrated by the GP inbred line (54 cm) for leaf length and the HW19 line (3.9 cm) for leaf width under CND treatments (Table 1). Another study demonstrated that ZnO treatment at 10 mg/L in rice greatly improved leaf length (33 mm) without affecting leaf width [51]. In salt-stressed rapeseed (200 mM NaCl, 12 days), 0.05 mM poly(acrylic) acid-coated nanoceria increased leaf width and length by 25% and 31%, respectively [52]. Furthermore, low doses of MnFe_2_O_4_ NP up to 250 mg/L gradually enhanced *Hordeum vulgare* leaf blade length [43]. In contrast, Lebedev et al. [53] found that different levels of nanoparticles (Fe^0^, Fe_3_O_4_, and FeSO_4_) inhibited leaf elongation in *Triticum vulgare* compared with untreated plants. This is consistent with the negative responses to nanoparticles of a few inbred lines in the current experiment. It is reported that larger leaf size might enhance plant photosynthesis and indirectly improve abiotic stress tolerance [54].

In plants with fibrous root systems, a longer root length can facilitate the absorption of water and nutrients from the extensive rhizosphere, enhancing the water status of the plants and increasing their productivity under drought stress [55]. The application of nitrogen-doped carbon nanodots (N-CD) at a dose of 5 mg L^−1^ substantially increased the root length of drought-stressed maize by 106.8% [31], which is consistent with the results in this study for the CND-treated HW9 line (Table 2). Wang et al. [56] reported that foliar application of CND (5 mg L^−1^) increased maize root length by 21.4%. Further, a study conducted on mung bean sprouts demonstrated a maximum increase of 29.9 and 18.3% in root and stem length, respectively, when treated with 0.02 mg mL^−1^ CND [57]. Su et al. [58] found that 180 mg L^−1^ CND enhanced the root and seedling length of peanut plants by 1.5 times over the control. Moreover, Yang et al. [44] demonstrated that drought-stressed maize roots responded positively to foliar CND application at 5 mg L^−1^ with an increase of 167.9% in root length. Meanwhile, Tombuloglu et al. [43] reported a steady increase in root and stem length of barley on exposure to MnFe_2_O_4_ NP up to 250 mg L^−1^, then a gradual decline. The root length of tomato plants increased by 53% when treated with 10 mg L^−1^ MnFe_2_O_4_ NP [41], which is lower than the increase observed in this study in the MnFe_2_O_4_ NP-treated 12BS5076-8 maize inbred line (Table 2). On the contrary, Cantu et al. [59] concluded that there was no statistical difference in root and shoot length between 250 mg/L MnFe_2_O_4_ NP-treated tomato plants and control plants, which also matches a few of the results of the current study (Table 2). The results from the previous studies mentioned above show a trend that is nearly identical to that observed in the current study, although the extent of increases or decreases compared with the control differ, which may be because of differences in growth conditions, genetic variations among crop varieties, nanoparticle concentrations, and application methods.

Drought disrupts the balance between water uptake from soil and its loss through transpiration, which adversely affects plant growth and development [60]. However, it has been reported that nano chemicals such as CND can notably improve the water absorption capacity of plants by enhancing their root activity under drought stress [61]. When drought-stressed tomato plants were treated with functional carbon nanodots (FCN) at 3 mg L^−1^, the leaf moisture content (LMC) of the plants was considerably higher than that of untreated drought-stressed plants [61], which was in agreement with the LMC results obtained for the majority of inbred lines in the current study (Figure 2a). Although the leaf relative water content (LRWC) of salt-stressed *Vigna radiata* increased with the application of trehalose and glucose-terminated carbon nanodots (CNPT and CNPG), the increase was not significant [30]. A 500 mg L^−1^ foliar application of Si-Zn NPs to soybean improved LRWC by a maximum of 9.5% under drought stress [62], which is less than the improvement observed in this study for the MnFe_2_O_4_ NP-treated HW15 line (57.4%) (Figure 2b). It was also found that 400 mg L^−1^ Cu_2_O and CuO nanoparticles decreased the RWC of cucumber leaves by 14.3% and 17%, respectively [63], and the LRWC results of the current study also indicated a similar effect in some maize-inbred lines (Figure 2b). When drought stress of 4% soil moisture content was applied, the water saturation deficit (WSD) in the leaves of different barley genotypes increased from 10.4 to 100.0% compared to controls [64]. Moreover, in drought-stressed mung bean genotypes, the water saturation deficit increased, ranging from 23.7 to 47.2% for most genotypes [65], which corroborates our WSD results (Figure 2c).

A comparative study on the morphology and application methods of zinc oxide nanoparticles (ZnO NPs) found that tomato stem diameter increased when hexagonal and spherical ZnO NPs were applied to plant foliage [66], which supports the nanoparticle application method used in the current study (Figure 3). According to Mazhar et al. [67], drought stress reduced the stem diameter of flax plants; however, treatment with different doses of iron oxide nanoparticles greatly augmented the stem diameter. Furthermore, it was also discovered that tomato seedling stem diameter was reduced at all evaluated carbon nanotube concentrations [68].

The fresh weight of maize shoots and roots increased by 232.5 and 140% on exposure to 5 mg L^−1^ of CND [31]. In comparison, the highest increases in maize shoot and root fresh weight under CND treatment in our study were 64.4 and 275.5%, respectively (Table 3). Another study conducted on maize reported that 5 mg L^−1^ CND increased the fresh weight of roots and shoots by 18.9 and 13.8%, respectively [56]. According to Chen et al., [69] tomato seedlings treated with 16 mg kg^−1^ FCN improved the fresh weight of their root and aerial parts by 124.5 and 35.7%, respectively, in saline-alkaline soil. It was also found that foliar CND application improved the fresh weight of maize roots and shoots by 50.6 and 62.1%, respectively, compared with the control [44]. Among MnFe_2_O_4_ NP treatments, barley showed 10.3% higher seedling fresh weight at 250 mg L^−1^ MnFe_2_O_4_ NP than the control [43]. Plant drought stress-alleviating effects of CND or MnFe_2_O_4_ NP nanomaterials may vary depending on several factors, including soil and environmental conditions, nanomaterial properties, plant species and their growth stages, and physiological characteristics.

The SPAD-502 chlorophyll meter has proven to be an effective instrument for fast and non-destructive estimation of plant total chlorophyll content, and leaf chlorophyll concentration is the most reliable indicator of plant photosynthetic activity [70]. Previously, Wang et al. [56] reported that 5 mg L^−1^ N-CD treatment improved maize leaf chlorophyll content by 15.4%, while the HCW1 line of this study exhibited the highest increase of 39.8% in chlorophyll content among the CND-treated group (Figure 4). Another study found a statistically significant (*p* ≤ 0.05) increase in the total chlorophyll content of tomato plants treated with FCN under drought stress [61]. The application of 16 mg kg^−1^ FCN to tomato seedlings under saline-alkali stress promoted the total chlorophyll content of leaves by 3.3 times compared with the control [69]. It has also been reported that the chlorophyll content of mung beans increased by 14.8% after CND treatment. On the other hand, Cantu et al. [59] found no significant differences in chlorophyll content between tomato plants treated with 250 mg L^−1^ MnFe_2_O_4_ NP and their control plants. Furthermore, when MnFe_2_O_4_ NP were applied to barley plants at concentrations ranging from 62.5 to 500 mg L^−1^, the results showed no significant differences in chlorophyll content [43]. However, another experiment conducted on tomato plants showed that MnFe_2_O_4_ NP-treated plants had 20% higher chlorophyll levels than untreated plants [41]. In the current experiment, the MnFe_2_O_4_ NP-treated HW-12 line exhibited the highest increase (64.6%) in chlorophyll content (Figure 4). Aside from improving chlorophyll content, CND can also increase the electron transfer rate, PSII functioning, and rubisco activity, resulting in enhanced photosystem activity and crop yield [57,71]. Furthermore, fluorescent nanomaterials were reported to enhance solar energy harvesting, thus facilitating the capture of chloroplast carbon, harnessing solar energy, and influencing the sensing processes [72].

The DPPH free radical scavenging assay is widely used to assess the antioxidant activity of plants and food items because of its high accuracy and simplicity [73]. As an organic free radical, DPPH^•^ is capable of absorption in the UV spectrum. When antioxidants containing plant extracts are added to the DPPH solution, the unpaired electrons in the DPPH^•^ are paired and reduced, resulting in a gradual fading of the dark purple color in the DPPH solution. Antioxidant activity is evaluated by measuring DPPH solution absorption [74]. Zahedi et al. [75] reported an increase of 11% in DPPH free radical scavenging when drought-stressed strawberry plants were treated with Se/SiO_2_-NP compared with untreated plants. It was discovered that TiO_2_ NP treatment of saffron increased the DPPH free radical scavenging potential by 9–26% [76]. In addition, tomato plants treated with 250 mg L^−1^ SiO_2_ NP showed 3.5% higher antioxidant activity in hydrophilic compounds than control plants [77]. A remarkable increase in DPPH free radical inhibition was observed in *Medicago sativa* leaves treated with 50 or 100 ppm TiO_2_ NM [78], supporting most of the DPPH results of the present study (Figure 5).

In plants subject to abiotic stress, phenolic compounds play a crucial role in protein synthesis, photosynthesis, allelopathy, and enzyme activity [79]. A phenolic compound acts as a nucleophile, reacting with oxygen radicals such as superoxide, hydroxyl ion, and lipid peroxyl radicals [80]. This inhibits lipid peroxidation by removing free radicals and preventing damage [81]. When *Brassica napus* leaves were treated with 100 µM melatonin under 300 µM cobalt stress, their TPC increased by 115% [82], which is in accordance with the TPC of MnFe_2_O_4_ NP-treated 17CS8067 line (115.2%) (Figure 6). The application of spermine (25 mg L^−1^), 24-epibrassinolide (0.1 mg L^−1^), and silicon (7 mg L^−1^) enhanced the phenolic contents in maize leaves by 45.1, 32.8, and 50.1%, respectively, under water stress conditions compared with the untreated group [83]. Another study reported a statistically significant (*p* ≤ 0.001) improvement in the TPC of maize plants exposed to drought stress. However, 6 mM silicon seed priming produced a statistically significant (*p* ≤ 0.001) decrease in the TPC of maize plants under well-watered and drought-stressed conditions [84]. Likewise, when salt-stressed maize plants were treated with biostimulants (Megafol–Meg), the TPC content decreased from 821 ± 102 to 697 ± 74 µg GAE/g samples in comparison with untreated stressed plants, which is in line with some of the current study TPC results (Figure 6).

Phenolic acids, a group of phenolic compounds, act in plants as secondary metabolites and have benzene rings with one or more hydroxyl groups [85]. They play a critical role in the plant’s resistance to pathogens and herbivores, plant growth regulation, and prevention of oxidative stress [86]. The phenolic acids in food plants occur as esters or glycosides conjugated with certain compounds such as sterols, flavonoids, glucosides, and hydroxyl fatty acids [85]. Based on structure, phenolic acids are categorized into hydroxybenzoic acids (HBAs) and hydroxycinnamic acids (HCAs). A majority of HBAs contain a C6-C1 backbone obtained directly from benzoic acid, and they include gallic acid, salicylic acid, etc., and HCAs consist of a C6-C3 phenylpropanoid structure and include ferulic acid, caffeic acid, *p*-coumaric acid, etc. [85]. Thus, this study examined the content of both HBAs (gallic acid and syringic acid) and HCAs (ferulic acid, caffeic acid, *p*-coumaric acid, and chlorogenic acid) available in drought-stressed maize inbred lines under different NP treatments (Figure 7). According to Kolo et al. [87], drought stress caused a 0.3-fold decrease of *p*-coumaric acid in maize leaf, whereas caffeic acid and ferulic acid in leaf increased by 0.9-fold and 0.3-fold, respectively. Further, Rayee et al. [88] examined 13 standard phenolic acids in MNR2 and Koshihikari rice varieties under chilling stress. However, only six (vanillin, sinapic acid, benzoic acid, ferulic acid, cinnamic acid, and ellagic acid) were detected in the leaves of the chilling stressed Koshihikari rice variety, and three (benzoic acid, ellagic acid, and cinnamic acid) in the MNR2 variety. There was an increase in gallic acid, chlorogenic acid, and *p*-coumaric acid contents in *Amaranthus tricolor* leaves following 25 mM NaCl stress. However, decreases in syringic acid, caffeic acid, and ferulic acid contents were observed [89]. In a study on *Amaranthus tricolor* under drought stress, HBAs were found to be the most abundant phenolic acids in this genotype. Among HBAs, salicylic acid was the predominant phenolic acid, followed by vanillic and gallic acids. Among HCAs, chlorogenic acid was the most prevalent phenolic acid, followed by *trans*-cinnamic and *m*-coumaric acids. In addition, considerable amounts of *p*-coumaric, caffeic, and ferulic acids were also identified [90]. Under different levels of salinity stress, caffeic acid, syringic acid, and salicylic acid as free phenolic acids were not detected in einkorn, durum wheat, and emmer sprouts. However, *p*-coumaric acid and *trans*-ferulic acid contents ranged from 4.1 to 10.9 and 7.2 to 21.9 µg g^−1^, respectively, among the three genotypes under salinity stress [91].

## 4. Conclusions

In summary, the inbred lines that demonstrated the highest values for leaf length, leaf width, leaf relative water content, root length, stem diameter, and root fresh weight belonged to the CND-treated group. Moreover, the accumulation of chlorogenic acid, caffeic acid, and syringic acid was recorded at their maximum levels under CND treatment. Conversely, the maximum shoot length, leaf moisture content, shoot fresh weight, leaf chlorophyll content, and DPPH free radical scavenging activity were observed in the respective MnFe_2_O_4_ NP-treated inbred lines. The differences in results among various inbred lines for the same or distinct phenotypic traits may be attributed to a variety of factors, including differences in the chemical properties of the applied nanoparticles, their compatibility level with specific lines for respective phenotypic traits, genetic divergence among the inbred lines, etc. To the best of our knowledge, this is the first study examining the effects of CND and MnFe_2_O_4_ NP on drought-stressed maize. Therefore, this study will help researchers to design further experiments and evaluate further in-depth the crosstalk between these biostimulants and drought stress in different crops.

## 5. Materials and Methods

### 5.1. Collection of Experimental Material

A total of 41 elite maize inbred lines (EMILs) were used in this study, and they were developed by and received from the Maize Experimental Station, Gangwon Agricultural Research and Extension Service, Hongcheon, South Korea. Most of these inbred lines were derived from waxy maize; however, some originated from flint and popcorn maize. The EMILs are used as parental lines for the development of numerous F1 hybrids (Appendix A).

This study employed two types of nanomaterials: manganese ferrite (MnFe_2_O_4_) nanoparticles (NP) and carbon nanodots (CND). Research-grade MnFe_2_O_4_ and CND were purchased from Nanografi Nanotechnology Company (Ankara, Turkey) and Ossila Limited (Sheffield, UK), respectively. The characterization information of the MnFe_2_O_4_ and CND by the respective company is expressed in Table 4.

### 5.2. Seedbed Media Characteristics

An artificial seedbed was created in pots using potting mix acquired from Seoul Bio Co., Ltd. (Seoul, Republic of Korea). The names of raw materials, their mixing ratios, and the physicochemical properties of the potting mix provided by the company are depicted in Table 5. The pots were evenly filled with potting mix, and each pot had nine holes at the bottom.

### 5.3. Experimental Design and Crop Husbandry

A pot study was conducted at the glass house of the College of Agriculture and Life Sciences, Kangwon National University, Gangwon-do, Republic of Korea (37°52′ N, 127°44′ E). A preliminary study was conducted to determine the optimal dose of CND and MnFe_2_O_4_ NP for boosting maize drought stress tolerance. In the preliminary study, five concentrations of CND (0, 5, 10, 20, and 40 mg L^−1^) and six concentrations of MnFe_2_O_4_ NP (0, 100, 200, 300, 400, and 500 mg L^−1^) were selected based on the published literature and sprayed on plant foliage grown under drought stress at 30% field capacity (FC) for five consecutive days. The drought stress condition (30% FC) was maintained by following the method described in [90] with a slight modification of adding evapotranspired water daily. Briefly, the gravimetric method was used to measure the field capacity of the potted soil used in the pots. Each pot was evenly filled with a weighed amount of completely dried potted soil. The amount of water required to maintain the respective FC (80 and 30%) was calculated from 100% FC, which is obtained by subtracting the dry soil weight from the weight of potted soil at 100% FC. Different morphological parameters were measured in conjunction with the plant pigment content (Appendix A). Based on the results of the preliminary study, 10 mg L^−1^ CND and 300 mg L^−1^ MnFe_2_O_4_ were selected for the principal experiment. In autumn 2022 (October–November), the main experiment was performed under the same drought condition (30% FC) as the preliminary experiment using three treatments: control (foliar spray of distilled water), foliar spray of CND (10 mg L^−1^), and foliar spray of MnFe_2_O_4_ (300 mg L^−1^). The glass house temperature was maintained between 26–31 °C during the day and 15–20 °C at night. As there were no restrictions on sunlight exposure, the photoperiod in the glass was naturally regulated by daylight. Maize plants were first grown under well-watered conditions (80% FC) until they reached the trifoliate stage. As the third leaf collar appeared, drought stress was imposed on plants by limiting additive water to 30% FC. The drought stress lasted until the plants were harvested, and foliar applications of selected nanoparticles began on the 8th day and were completed on the 12th day of drought stress. During these five consecutive days of foliar sprays, each plant group received 5ml of its respective aqueous nanoparticle solution per day, while the control group received the same quantity of distilled water. Plants were harvested on the 21st day of drought stress to measure various growth and stress-related parameters. This study was carried out in a completely randomized design (CRD). Each treatment was replicated three times. All agronomic practices were uniform except for the factors under study.

### 5.4. Data Collection

The maize genotypes used in the experiment are inbred lines; therefore, it was critical to test their germination percentage (GP) under normal conditions. A count of germinated seeds was initiated once 50% of the seed germination was complete, and plumules continued to be counted until the number was constant. GP was calculated by using the following formula:(1)GP=seeds germinated÷total seeds sown×100

Plant morphological parameters were measured immediately following the harvesting of plants from each treatment group. Plant root and shoot lengths were recorded using a meter rod. Root and shoot fresh weights were measured on a digital weighing balance (AG204, Mettler Toledo Ltd., Greifensee, Switzerland). Plant stem diameter was measured using a dial caliper (Mitutoyo Corporation, Kawasaki, Japan). Leaf blade length and width were measured from the third fully grown leaf using a ruler.

Leaf moisture content (*LMC*) was estimated by using the following equation [92]:(2)$$LMC\;\left(\%\right)=\frac{LFW-LDW}{LFW}\times 100$$

Leaf relative water content (*LRWC*) and water saturation deficit (*WSD*) was measured using the following equations [93]:(3)$$LRWC\;\left(\%\right)=\frac{LFW-LDW}{LTW-LDW}\times 100$$
(4)$$WSD=\frac{LTW-LFW}{LTW-LDW}\times 100$$
where *LFW* is the leaf’s fresh weight, *LDW* is the leaf’s dry weight, and *LTW* is the leaf’s turgid weight. *LDW* was measured after over-drying at 105 °C until constant weight, and *LTW* was measured by softly wiping the soaked leaves in distilled water for 12 h at room temperature.

SPAD (Soil Plant Analysis Development) chlorophyll content was estimated by using a SPAD meter (SPAD-502, Konica Minolta, Tokyo, Japan) [94]. Measurements were done in the morning, just before the harvesting of plants. For each replication of treatments, three readings were recorded to obtain the average value.

The free radical scavenging activity of 2,2-diphenyl-1-picrylhydrazyl (DPPH) was estimated by following the method used by Choi et al. [95] with slight modifications. The sample for analysis was prepared by adding 4 mL of pure methanol (MeOH) to 0.1 g of dried plant powder and diluting it 10 times. Then 0.1 mL of the diluted sample was mixed with 0.1 mL of a 0.15 mM DPPH solution in 96-well plates, and the reaction was allowed to proceed for 30 min in the dark. The absorbance was measured by ELISA (model 680, Bio-Rad Inc., Hercules, CA, USA) at 515 nm to check the DPPH radical scavenging activity by using the following equation [73]:(5)$$DPPH\;\;free\;radical\;scavenging\;potential\;\left(\%\right)=[1-\left(AbS-AbC\right)]\times 100$$
where *AbS* is the absorbance of the test sample and *AbC* is the absorbance of the control.

Total phenolic content (TPC) was measured using the Folin-Ciocalteau reagent method [96] with minor changes. A methanol-extracted sample (0.1 mL) was mixed with 0.05 mL of Folin-Ciocalteau reagent, 0.3 mL of 20% sodium carbonate, and 1 mL of distilled water. The mixture was allowed to react for 20 min at room temperature. The absorbance was measured at 725 nm against a blank sample using an ultraviolet (UV)/visible light (VIS) spectrophotometer (V530, Jasco Co., Tokyo, Japan). The TPC was calculated by preparing a standard calibration curve using standard gallic acid solutions in the range of 10–250 µg/mL, and TPC content was expressed as gallic acid equivalent in mg per g of sample (mg GAE/g sample).

Analysis of phenolic compounds was carried out using high-performance liquid chromatography-UV (HPLC-UV) analysis [97]. Briefly, the HPLC analysis was performed using an Agilent 1260 series instrument and a Shiseido (Tokyo, Japan) Capcell Pak C18 column. A series of phenolic compounds were measured using their corresponding standard solutions: gallic acid, chlorogenic acid, *p*-coumaric acid, ferulic acid, syringic acid, and caffeic acid. The mobile phase consisted of 0.1% phosphoric acid in water and acetonitrile, and the flow rate was set at 1ml/min. The UV detector was adjusted to a wavelength of 270 nm. Each phenolic compound was determined by comparing its retention time with the respective standard under the same conditions. Quantification of the phenolic compounds was performed by using standard curves (0.5, 1, 5, 10, 25, 50, and 100 ppm) with external standards. The results were stated as µg/mL.

### 5.5. Statistical Analysis

An analysis of variance (one-way ANOVA) was performed on the collected data using IBM SPSS Statistics version 26 (IBM Corp., Armonk, NY, USA). In addition, Duncan’s post hoc test (*p* ≤ 0.05) was performed to separate the means. The results are presented as the means ± SD (standard deviation) of three replications. Graphical presentation of data was done using Microsoft Excel 365 (Version 2303) and TBtools software [98].

## Figures and Tables

**Figure 1 plants-12-02922-f001:**
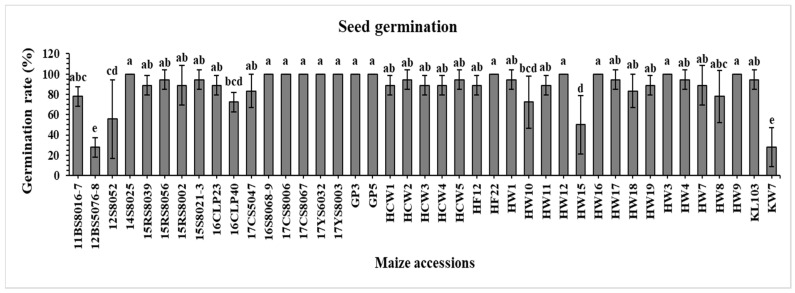
Germination percentage of 41 maize inbred lines under well-watered conditions. The results are expressed as the means ± standard deviation (*n* = 3). Different lowercase letters indicate significant differences among different cultivars at *p* ≤ 0.05.

**Figure 2 plants-12-02922-f002:**
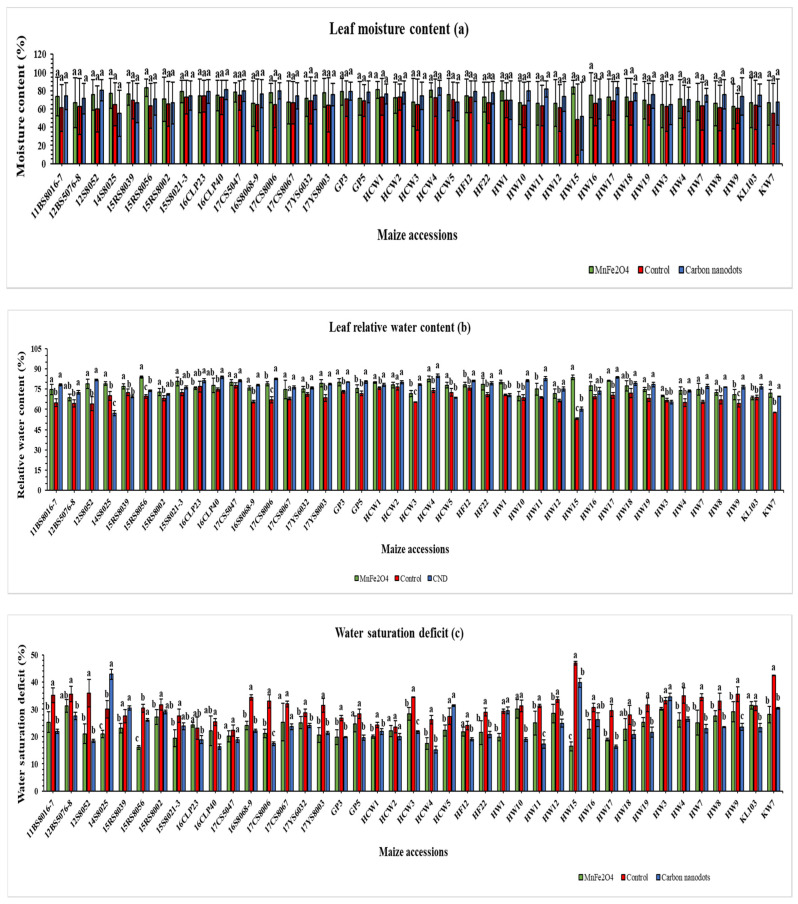
Effects of CND and MnFe_2_O_4_ NP on leaf moisture content (**a**), leaf relative water content (**b**), and water saturation deficit (**c**) of 41 maize inbred lines under drought stress. The results are expressed as the means ± standard deviation (*n* = 3). Different lowercase letters within the bars of each inbred line indicate statistically significant differences among applied treatments on that line at *p* ≤ 0.05.

**Figure 3 plants-12-02922-f003:**
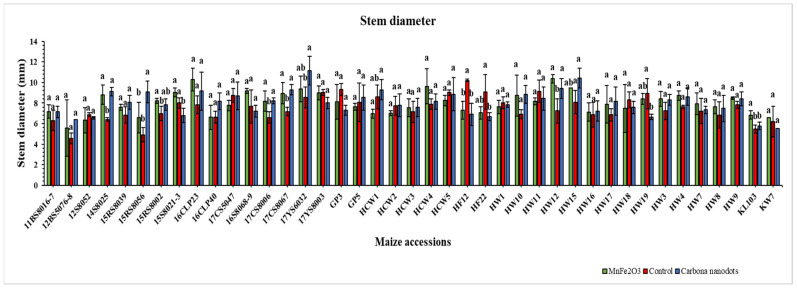
Effects of CND and MnFe_2_O_4_ NP on stem diameter of 41 maize inbred lines under drought stress. The results are expressed as the means ± standard deviation (*n* = 3). Different lowercase letters within the bars of each inbred line indicate statistically significant differences among applied treatments on that line at *p* ≤ 0.05.

**Figure 4 plants-12-02922-f004:**
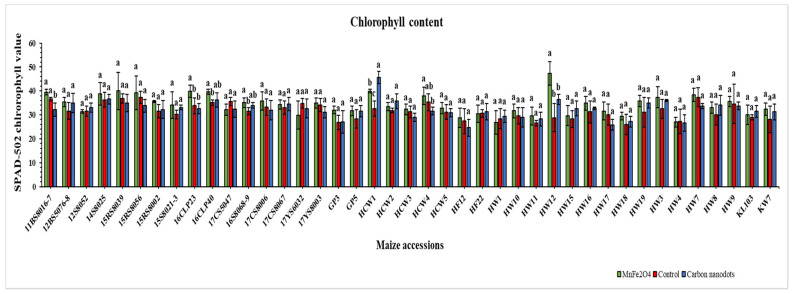
Effects of CND and MnFe_2_O_4_ NP on leaf chlorophyll content of 41 maize inbred lines under drought stress. The results are expressed as the means ± standard deviation (*n* = 3). Different lowercase letters within the bars of each inbred line indicate statistically significant differences among applied treatments on that line at *p* ≤ 0.05.

**Figure 5 plants-12-02922-f005:**
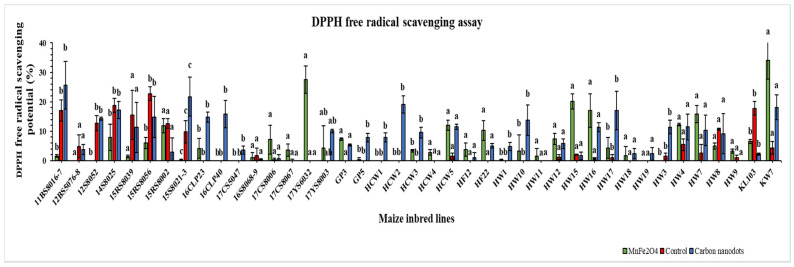
Effects of CND and MnFe_2_O_4_ NP on DPPH free radical scavenging activity of 41 maize inbred lines under drought stress. The results are expressed as the means ± standard deviation (*n* = 3). Different lowercase letters within the bars of each inbred line indicate statistically significant differences among applied treatments on that line at *p* ≤ 0.05.

**Figure 6 plants-12-02922-f006:**
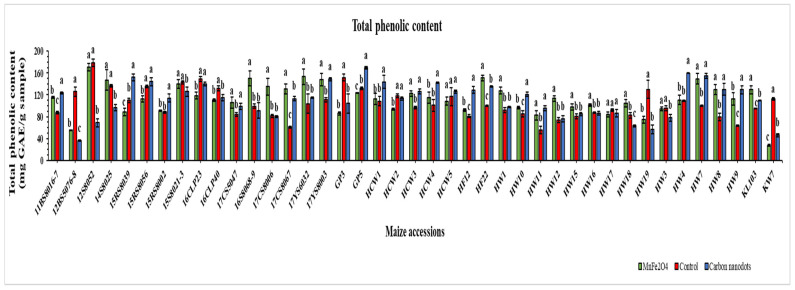
Effects of CND and MnFe_2_O_4_ NP on total phenolic content of 41 maize inbred lines under drought stress. GAE: Gallic acid equivalent. The results are expressed as the means ± standard deviation (*n* = 3). Different lowercase letters indicate statistically significant differences among applied treatments on each line at *p* ≤ 0.05.

**Figure 7 plants-12-02922-f007:**
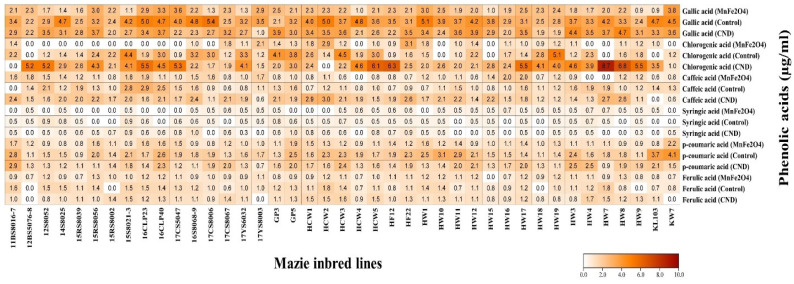
Effects of CND and MnFe_2_O_4_ NP on concentrations of different phenolic acids of 41 maize inbred lines under drought stress. The results are presented as the means of three replications.

**Table 1 plants-12-02922-t001:** Effects of CND and MnFe_2_O_4_ NP on leaf blade length and width of 41 maize inbred lines under drought stress.

Maize Accessions	Leaf Blade Length (cm)			Leaf Width (cm)		
	MnFe_2_O_4_	Control	CND	MnFe_2_O_4_	Control	CND
11BS8016-7	42.4 ± 1.4 b	36.4 ± 2.1 c	48.6 ± 2.8 a	2.3 ± 0.3 a	2.2 ± 0.4 a	2.8 ± 0.7 a
12BS5076-8	36.9 ± 6.1 a	29.5 ± 1.9 b	32.7 ± 0.0 ab	2.0 ± 0.3 b	1.8 ± 0.2 b	2.3 ± 0.0 a
12S8052	40.7 ± 2.6 b	36.3 ± 1.2 b	47.8 ± 5.2 a	2.1 ± 0.3 b	2.4 ± 0.4 ab	2.8 ± 0.2 a
14S8025	40.1 ± 3.9 a	38.1 ± 1.7 a	36.4 ± 0.9 a	3.0 ± 0.3 a	3.1 ± 0.4 a	2.3 ± 0.4 b
15RS8039	32.9 ± 2.6 ab	27.9 ± 3.5 b	35.2 ± 2.9 a	2.2 ± 0.4 a	2.4 ± 0.3 a	2.4 ± 0.6 a
15RS8056	29.3 ± 2.5 b	31.7 ± 2.1 b	43.2 ± 3.9 a	2.3 ± 0.3 a	2.0 ± 0.3 a	2.1 ± 0.3 a
15RS8002	38.9 ± 2.7 a	32.6 ± 3.0 ab	31.4 ± 4.1 b	3.0 ± 0.3 a	2.0 ± 0.5 a	2.4 ± 0.8 a
15S8021-3	38.7 ± 2.8 a	38.7 ± 1.7 a	38.9 ± 2.1 a	2.4 ± 0.4 a	2.3 ± 0.5 a	2.1 ± 0.7 a
16CLP23	32.7 ± 0.8 b	43.9 ± 4.2 a	48.2 ± 2.3 a	2.3 ± 0.6 a	2.6 ± 0.5 a	3.0 ± 0.2 a
16CLP40	34.2 ± 3.1 b	39.8 ± 2.4 ab	44.3 ± 3.6 a	3.4 ± 0.5 a	2.9 ± 0.4 a	3.3 ± 0.6 a
17CS5047	33.4 ± 2.2 b	44.7 ± 2.2 a	45.7 ± 3.3 a	2.9 ± 0.4 a	3.0 ± 0.7 a	2.8 ± 0.9 a
16S8068-9	34.4 ± 2.6 b	41.2 ± 1.8 a	32.1 ± 1.8 b	3.1 ± 0.4 a	3.2 ± 0.4 a	3.2 ± 0.4 a
17CS8006	42.0 ± 3.3 a	42.2 ± 3.3 a	36.6 ± 1.9 a	3.9 ± 0.3 a	3.7 ± 0.4 a	2.8 ± 0.6 b
17CS8067	39.5 ± 2.8 a	29.8 ± 2.2 b	41.4 ± 3.0 a	2.2 ± 0.4 a	2.6 ± 0.3 a	2.3 ± 0.4 a
17YS6032	42.0 ± 3.1 a	43.4 ± 2.0 a	45.7 ± 2.8 a	3.2 ± 0.4 a	2.9 ± 0.4 a	3.5 ± 0.5 a
17YS8003	33.2 ± 2.5 ab	35.1 ± 2.9 a	28.5 ± 3.4 b	2.9 ± 0.3 a	3.1 ± 0.4 a	2.7 ± 0.6 a
GP3	46.4 ± 1.9 a	43.0 ± 3.2 a	46.5 ± 3.7 a	3.0 ± 0.5 a	2.5 ± 0.6 a	2.4 ± 0.6 a
GP5	49.6 ± 7.9 a	43.8 ± 3.9 a	54.0 ± 2.4 a	3.4 ± 0.5 a	2.9 ± 0.8 a	3.0 ± 0.8 a
HCW1	38.0 ± 5.4 a	33.4 ± 2.0 a	35.3 ± 3.5 a	3.0 ± 0.5 a	2.0 ± 0.5 b	3.0 ± 0.3 a
HCW2	40.5 ± 3.9 a	37.6 ± 2.4 a	44.3 ± 4.3 a	2.7 ± 0.4 a	2.7 ± 0.7 a	3.2 ± 0.9 a
HCW3	35.7 ± 2.8 a	43.9 ± 3.9 b	45.8 ± 2.4 b	2.5 ± 0.4 b	2.3 ± 0.7 b	3.4 ± 0.2 a
HCW4	42.2 ± 4.3 a	39.5 ± 3.2 a	46.0 ± 4.2 a	2.9 ± 0.4 a	2.9 ± 0.6 a	3.3 ± 0.6 a
HCW5	35.3 ± 5.8 a	37.8 ± 2.4 a	38.8 ± 1.2 a	3.9 ± 0.4 a	3.1 ± 0.6 a	2.9 ± 0.8 a
HF12	47.2 ± 4.8 a	48.3 ± 2.7 a	47.1 ± 4.4 a	3.2 ± 0.5 a	2.2 ± 0.2 a	2.8 ± 0.8 a
HF22	40.3 ± 2.0 ab	36.9 ± 2.2 b	45.7 ± 5.5 a	3.4 ± 0.5 a	2.5 ± 0.4 b	3.3 ± 0.4 ab
HW1	41.4 ± 3.3 a	42.4 ± 1.5 a	41.2 ± 4.4 a	3.3 ± 0.7 a	2.6 ± 0.7 a	3.1 ± 0.3 a
HW10	48.7 ± 5.4 a	43.5 ± 2.0 a	49.8 ± 1.5 a	2.8 ± 0.6 a	2.6 ± 0.2 a	3.6 ± 0.7 a
HW11	38.8 ± 4.3 ab	43.7 ± 3.7 a	36.1 ± 2.1 b	2.6 ± 0.3 a	2.9 ± 0.7 a	3.4 ± 0.6 a
HW12	48.2 ± 2.8 a	45.6 ± 4.0 a	49.0 ± 4.2 a	2.9 ± 0.3 a	2.8 ± 0.4 a	3.1 ± 0.4 a
HW15	48.8 ± 0.0 a	46.5 ± 1.9 a	39.9 ± 3.3 b	3.2 ± 0.0 a	2.7 ± 0.5 a	3.0 ± 0.8 a
HW16	36.5 ± 1.9 b	36.7 ± 2.2 b	43.5 ± 4.3 a	2.3 ± 0.6 a	2.0 ± 0.4 a	2.0 ± 0.4 a
HW17	36.5 ± 1.3 b	44.3 ± 3.4 a	40.2 ± 1.4 ab	1.9 ± 0.8 a	2.9 ± 0.9 a	2.5 ± 0.9 a
HW18	35.9 ± 1.4 b	44.9 ± 1.1 a	34.5 ± 3.2 b	2.7 ± 0.2 a	3.3 ± 0.5 a	3.1 ± 0.5 a
HW19	41.4 ± 2.2 a	41.3 ± 2.2 a	35.3 ± 3.4 b	3.6 ± 0.5 ab	3.1 ± 0.2 b	3.9 ± 0.4 a
HW3	42.5 ± 2.8 a	41.6 ± 3.7 a	44.9 ± 3.8 a	3.8 ± 0.7 a	1.6 ± 0.3 b	3.2 ± 0.4 a
HW4	42.2 ± 6.0 a	49.3 ± 2.8 a	46.5 ± 3.2 a	2.2 ± 0.4 b	2.1 ± 0.4 b	3.1 ± 0.5 a
HW7	42.8 ± 1.3 a	44.0 ± 3.2 a	39.2 ± 2.4 a	2.7 ± 0.6 a	2.8 ± 0.4 a	3.0 ± 0.8 a
HW8	32.5 ± 2.7 b	28.4 ± 1.6 b	37.8 ± 3.2 a	2.7 ± 1.0 a	2.1 ± 0.5 a	3.2 ± 0.5 a
HW9	38.7 ± 2.4 a	33.5 ± 3.4 a	36.3 ± 3.4 a	2.9 ± 0.4 a	2.5 ± 0.3 a	2.4 ± 0.4 a
KL103	43.0 ± 3.7 a	33.5 ± 2.8 b	36.6 ± 4.5 ab	2.5 ± 0.6 a	2.5 ± 0.4 a	2.2 ± 0.4 a
KW7	36.8 ± 0.0 a	37.1 ± 2.1 a	34.3 ± 0.0 b	2.4 ± 0.0 a	1.9 ± 0.4 b	2.7 ± 0.0 a

Results are presented as the means ± standard deviation (*n* = 3). Different lowercase letters within each result indicate statistically significant differences among applied treatments on each line at *p* ≤ 0.05.

**Table 2 plants-12-02922-t002:** Effects of CND and MnFe_2_O_4_ NP on shoot and root length of 41 maize inbred lines under drought stress.

Maize Accessions	Shoot Length (cm)			Root Length (cm)		
	MnFe_2_O_4_	Control	CND	MnFe_2_O_4_	Control	CND
11BS8016-7	52.9 ± 0.2 b	52.4 ± 0.7 b	57.0 ± 2.3 a	18.4 ± 1.5 ab	15.8 ± 1.6 b	21.9 ± 4.2 a
12BS5076-8	48.2 ± 14.1 a	32.0 ± 2.5 b	34.1 ± 0.0 b	23.7 ± 1.1 a	8.6 ± 0.8 c	16.2 ± 0.0 b
12S8052	54.5 ± 0.9 b	58.2 ± 1.6 b	65.1 ± 4.8 a	20.8 ± 3.3 b	22.4 ± 0.4 ab	28.1 ± 6.9 a
14S8025	52.7 ± 3.0 a	45.1 ± 2.9 b	52.3 ± 2.7 a	18.4 ± 2.3 a	20.0 ± 4.1 a	20.6 ± 2.0 a
15RS8039	50.5 ± 3.7 b	41.7 ± 1.7 c	55.8 ± 1.0 a	30.9 ± 6.4 a	28.0 ± 4.6 a	24.0 ± 3.9 a
15RS8056	40.5 ± 1.1 b	39.8 ± 2.7 b	53.7 ± 1.6 a	17.6 ± 2.4 b	22.4 ± 1.2 b	36.9 ± 9.4 a
15RS8002	48.0 ± 9.1 a	43.3 ± 1.3 a	54.6 ± 4.4 a	24.7 ± 1.7 a	25.1 ± 3.3 a	24.5 ± 3.2 a
15S8021-3	54.0 ± 3.6 a	42.6 ± 2.6 b	55.3 ± 0.3 a	33.4 ± 4.9 a	22.2 ± 3.5 b	18.9 ± 1.6 b
16CLP23	51.2 ± 1.6 a	59.0 ± 0.3 a	56.2 ± 7.3 a	32.2 ± 3.8 ab	30.6 ± 2.3 b	37.8 ± 2.9 a
16CLP40	55.6 ± 5.5 a	55.2 ± 3.4 a	58.9 ± 2.2 a	37.0 ± 3.0 a	30.6 ± 1.9 b	37.8 ± 3.1 a
17CS5047	52.1 ± 2.1 a	53.8 ± 2.6 a	48.2 ± 10.8 a	17.6 ± 3.1 a	21.1 ± 3.4 a	23.6 ± 4.6 a
16S8068-9	54.1 ± 2.1 a	46.9 ± 5.9 a	48.4 ± 5.4 a	29.2 ± 1.7 b	36.9 ± 2.6 a	40.9 ± 3.7 a
17CS8006	44.0 ± 1.2 a	55.0 ± 1.8 a	55.7 ± 10.4 a	30.1 ± 11.0 a	22.9 ± 7.8 a	24.7 ± 6.1 a
17CS8067	55.1 ± 1.6 a	44.6 ± 3.3 b	57.6 ± 2.2 a	24.9 ± 3.9 b	22.4 ± 1.0 b	32.5 ± 2.8 a
17YS6032	56.7 ± 2.7 a	52.0 ± 2.9 a	56.7 ± 6.2 a	24.3 ± 2.1 ab	30.4 ± 0.9 a	20.7 ± 5.3 b
17YS8003	41.4 ± 3.4 a	47.7 ± 10.5 a	42.9 ± 6.6 a	22.1 ± 2.5 a	14.5 ± 2.9 a	19.6 ± 5.1 a
GP3	61.8 ± 1.7 a	55.8 ± 3.6 a	58.3 ± 5.5 a	16.8 ± 1.0 b	27.4 ± 3.9 a	26.5 ± 3.9 a
GP5	60.8 ± 1.8 a	61.5 ± 2.2 a	65.1 ± 2.4 a	21.1 ± 1.9 a	20.3 ± 3.4 a	21.3 ± 5.8 a
HCW1	55.8 ± 5.6 a	47.7 ± 5.6 a	51.7 ± 3.3 a	34.1 ± 0.3 a	25.3 ± 6.3 a	24.4 ± 6.8 a
HCW2	53.7 ± 1.2 a	50.4 ± 2.1 a	51.9 ± 6.6 a	32.6 ± 1.8 ab	26.0 ± 8.7 b	45.2 ± 7.8 a
HCW3	54.4 ± 4.1 a	55.8 ± 2.9 a	62.2 ± 5.8 a	24.8 ± 6.1 a	24.3 ± 5.3 a	30.1 ± 2.2 a
HCW4	52.8 ± 0.8 a	55.4 ± 3.5 a	55.5 ± 2.6 a	33.4 ± 9.8 a	23.4 ± 4.3 a	34.9 ± 4.8 a
HCW5	46.0 ± 0.7 a	54.6 ± 4.0 b	55.6 ± 1.9 b	25.9 ± 2.4 a	24.1 ± 6.1 a	20.4 ± 1.0 a
HF12	71.5 ± 0.6 a	61.5 ± 2.7 c	67.5 ± 2.0 b	23.0 ± 3.2 a	17.1 ± 1.1 b	23.1 ± 3.2 a
HF22	55.6 ± 1.8 a	55.2 ± 2.7 a	57.9 ± 4.8 a	23.7 ± 5.8 a	25.8 ± 3.7 a	20.7 ± 7.7 a
HW1	54.3 ± 0.6 ab	52.5 ± 2.1 b	58.9 ± 4.2 a	31.3 ± 15.8 a	20.2 ± 2.2 a	15.1 ± 3.7 a
HW10	59.5 ± 0.9 b	56.2 ± 1.0 c	63.9 ± 2.4 a	21.7 ± 4.3 a	27.2 ± 7.9 a	29.8 ± 6.5 a
HW11	59.6 ± 2.1 a	60.4 ± 2.7 a	56.6 ± 3.8 a	26.8 ± 4.2 a	23.7 ± 5.6 a	22.9 ± 6.4 a
HW12	55.8 ± 2.4 a	50.8 ± 14.6 a	58.4 ± 2.7 a	34.7 ± 2.6 a	23.2 ± 5.8 b	21.4 ± 3.5 b
HW15	57.4 ± 0.0 a	51.8 ± 6.0 a	57.7 ± 2.9 a	31.6 ± 0.0 a	20.3 ± 4.3 b	15.5 ± 2.2 b
HW16	41.9 ± 1.6 ab	40.9 ± 3.4 b	48.5 ± 4.5 a	25.3 ± 1.6 a	30.9 ± 7.7 a	25.6 ± 3.5 a
HW17	53.9 ± 5.7 a	54.5 ± 6.1 a	61.4 ± 1.9 a	25.1 ± 4.7 a	17.6 ± 1.7 a	22.7 ± 4.2 a
HW18	50.3 ± 10.8 a	56.4 ± 3.1 a	60.6 ± 1.6 a	22.1 ± 3.3 a	24.1 ± 2.2 a	18.2 ± 3.1 a
HW19	43.8 ± 6.7 a	50.7 ± 6.2 a	51.4 ± 1.0 a	19.7 ± 2.9 a	26.8 ± 2.6 a	21.9 ± 5.6 a
HW3	66.8 ± 2.9 a	52.6 ± 4.0 b	56.7 ± 5.3 b	22.9 ± 2.0 a	17.9 ± 2.7 a	21.8 ± 5.4 a
HW4	53.1 ± 1.3 ab	50.4 ± 6.5 b	59.9 ± 3.4 a	25.8 ± 8.7 a	20.4 ± 3.2 a	23.2 ± 6.2 a
HW7	52.1 ± 1.8 a	45.8 ± 5.2 a	47.8 ± 2.9 a	23.0 ± 2.6 a	27.4 ± 8.2 a	26.8 ± 9.3 a
HW8	44.2 ± 3.1 a	50.4 ± 3.9 a	48.8 ± 3.4 a	21.4 ± 6.3 a	15.9 ± 1.3 a	14.3 ± 3.0 a
HW9	55.8 ± 2.6 a	52.4 ± 3.9 a	51.7 ± 9.4 a	28.2 ± 4.6 ab	18.5 ± 3.2 b	37.6 ± 7.9 a
KL103	51.2 ± 2.8 a	40.1 ± 1.4 a	45.1 ± 10.8 a	14.7 ± 1.9 b	18.2 ± 2.9 ab	28.3 ± 9.9 a
KW7	53.4 ± 0.0 a	54.5 ± 4.1 a	47.6 ± 0.0 b	6.5 ± 0.0 b	19.5 ± 1.8 b	22.0 ± 0.0 a

Results are presented as the means ± standard deviation (*n* = 3). Different lowercase letters within each result indicate statistically significant differences among applied treatments on each line at *p* ≤ 0.05.

**Table 3 plants-12-02922-t003:** Effects of CND and MnFe_2_O_4_ NP on root and shoot fresh weight of 41 maize inbred lines under drought stress.

Maize Accessions	Root Fresh Weight (g)			Shoot Fresh Weight (g)		
	MnFe_2_O_4_	Control	CND	MnFe_2_O_4_	Control	CND
11BS8016-7	2.0 ± 0.5 b	3.0 ± 0.5 a	3.8 ± 0.3 a	7.8 ± 0.5 a	8.5 ± 0.8 a	8.9 ± 0.6 a
12BS5076-8	1.5 ± 0.4 b	1.6 ± 0.5 b	6.1 ± 0.0 a	6.6 ± 1.3 a	4.7 ± 0.5 b	7.1 ± 0.0 a
12S8052	2.9 ± 0.6 ab	2.3 ± 0.3 b	3.4 ± 0.2 a	6.2 ± 0.3 b	8.6 ± 0.5 a	8.7 ± 0.8 a
14S8025	4.2 ± 0.4 a	4.5 ± 0.4 a	3.1 ± 0.3 b	9.9 ± 0.4 a	9.1 ± 0.4 ab	8.3 ± 0.7 b
15RS8039	4.3 ± 0.5 a	4.7 ± 0.6 a	4.0 ± 0.4 a	10.3 ± 1.8 a	7.9 ± 0.3 b	7.0 ± 0.5 b
15RS8056	2.2 ± 0.6 b	1.1 ± 0.2 c	4.3 ± 0.4 a	5.4 ± 0.7 c	7.1 ± 0.2 b	8.6 ± 0.4 a
15RS8002	3.8 ± 0.9 b	4.3 ± 0.4 b	6.0 ± 0.3 a	11.2 ± 1.4 a	10.0 ± 0.3 a	11.3 ± 0.5 a
15S8021-3	3.2 ± 0.3 a	2.1 ± 0.4 b	2.2 ± 0.3 b	9.3 ± 0.8 a	7.6 ± 0.3 b	9.3 ± 0.6 a
16CLP23	4.9 ± 0.6 a	5.5 ± 0.9 a	3.9 ± 0.8 a	10.6 ± 1.1 a	7.9 ± 0.4 b	8.6 ± 1.3 b
16CLP40	3.4 ± 0.4 b	3.1 ± 0.4 b	5.1 ± 0.6 a	9.5 ± 0.9 a	7.5 ± 0.3 b	8.8 ± 0.9 ab
17CS5047	3.1 ± 0.5 a	3.8 ± 0.8 a	2.8 ± 0.6 a	6.1 ± 0.7 b	7.4 ± 0.8 a	5.9 ± 0.1 b
16S8068-9	3.3 ± 0.5 b	2.2 ± 0.4 c	4.7 ± 0.6 a	11.4 ± 1.9 a	10.6 ± 0.5 a	9.2 ± 1.0 a
17CS8006	2.4 ± 0.3 b	2.8 ± 0.3 b	4.1 ± 0.9 a	9.2 ± 0.3 ab	10.0 ± 0.9 a	8.3 ± 0.9 b
17CS8067	3.0 ± 1.0 a	2.4 ± 0.7 a	3.4 ± 1.2 a	8.0 ± 0.5 b	12.1 ± 0.3 a	7.5 ± 0.8 b
17YS6032	5.8 ± 0.5 a	5.7 ± 0.5 a	5.8 ± 0.4 a	12.2 ± 0.9 a	11.1 ± 0.5 ab	10.1 ± 0.6 b
17YS8003	3.6 ± 0.4 b	2.8 ± 0.3 b	6.8 ± 0.6 a	8.9 ± 1.2 a	5.9 ± 0.2 b	7.5 ± 1.0 ab
GP3	4.3 ± 0.3 b	3.8 ± 0.4 b	5.8 ± 0.7 a	10.5 ± 0.7 a	10.7 ± 0.5 a	10.1 ± 0.6 a
GP5	4.1 ± 0.4 b	5.0 ± 0.5 ab	5.9 ± 0.8 a	9.4 ± 1.0 a	7.1 ± 0.8 b	9.1 ± 0.2 a
HCW1	3.6 ± 0.3 b	3.8 ± 0.6 b	5.1 ± 0.6 a	9.8 ± 1.3 a	8.3 ± 0.1 a	9.0 ± 0.8 a
HCW2	2.8 ± 0.7 a	3.8 ± 0.7 a	4.0 ± 0.8 a	10.3 ± 0.7 a	9.3 ± 0.7 ab	8.4 ± 0.3 b
HCW3	3.2 ± 0.3 c	5.3 ± 0.4 b	7.0 ± 0.4 a	10.0 ± 1.0 a	9.2 ± 0.6 a	9.2 ± 0.6 a
HCW4	3.9 ± 0.2 b	3.6 ± 0.3 b	6.0 ± 0.9 a	10.5 ± 1.2 a	9.1 ± 0.4 ab	8.2 ± 1.0 b
HCW5	4.1 ± 0.5 b	5.3 ± 0.8 a	2.6 ± 0.4 c	6.0 ± 0.3 b	8.5 ± 0.5 a	8.4 ± 0.3 a
HF12	3.1 ± 0.7 a	3.8 ± 0.7 a	3.7 ± 0.7 a	8.6 ± 0.7 c	11.6 ± 0.3 a	10.0 ± 0.7 b
HF22	3.2 ± 0.2 c	4.3 ± 0.4 b	5.4 ± 0.3 a	9.9 ± 0.6 a	9.9 ± 0.6 a	9.0 ± 0.2 a
HW1	4.7 ± 0.5 a	2.6 ± 0.3 b	2.3 ± 0.6 b	11.4 ± 1.0 a	8.6 ± 0.2 b	8.9 ± 0.8 b
HW10	4.1 ± 0.5 b	6.1 ± 0.4 a	2.4 ± 0.6 c	7.1 ± 0.7 b	7.2 ± 0.4 b	10.6 ± 0.4 a
HW11	5.2 ± 0.3 a	5.0 ± 0.5 a	4.6 ± 0.4 a	7.2 ± 1.1 b	11.7 ± 0.4 a	7.1 ± 0.2 b
HW12	4.2 ± 0.4 a	4.3 ± 0.7 a	5.1 ± 0.3 a	11.9 ± 0.4 a	12.3 ± 0.6 a	8.2 ± 0.6 b
HW15	4.1 ± 0.0 ab	3.4 ± 0.6 b	4.4 ± 0.5 a	12.5 ± 0.0 a	7.9 ± 0.3 b	8.1 ± 0.5 b
HW16	3.7 ± 0.2 a	5.3 ± 0.3 a	4.0 ± 1.4 a	7.5 ± 0.5 b	7.6 ± 0.5 b	9.0 ± 0.2 a
HW17	4.5 ± 0.4 a	4.3 ± 0.5 a	5.0 ± 0.7 a	9.2 ± 0.6 b	10.1 ± 0.3 a	8.4 ± 0.2 b
HW18	3.8 ± 0.4 ab	3.9 ± 0.5 a	2.8 ± 0.6 b	6.3 ± 0.6 b	9.8 ± 1.0 a	10.4 ± 0.5 a
HW19	3.0 ± 0.3 b	5.8 ± 0.5 a	3.4 ± 0.9 b	8.3 ± 1.7 a	8.1 ± 0.4 a	6.8 ± 0.5 a
HW3	4.4 ± 0.8 a	5.3 ± 0.6 a	4.8 ± 0.9 a	9.2 ± 0.6 a	8.8 ± 0.4 a	9.0 ± 0.6 a
HW4	4.2 ± 0.4 b	6.0 ± 0.3 a	4.9 ± 0.4 b	9.4 ± 0.9 a	7.7 ± 0.5 b	9.3 ± 0.3 a
HW7	5.0 ± 0.5 a	4.0 ± 0.9 a	5.3 ± 0.6 a	10.9 ± 0.7 a	6.8 ± 0.2 b	7.0 ± 0.2 b
HW8	4.7 ± 0.3 a	2.3 ± 0.5 b	5.2 ± 0.4 a	8.0 ± 0.2 a	7.0 ± 0.3 b	8.4 ± 0.2 a
HW9	4.6 ± 0.4 a	2.3 ± 0.3 b	5.1 ± 0.8 a	10.5 ± 1.3 a	8.5 ± 0.7 b	8.0 ± 0.5 b
KL103	3.1 ± 0.3 a	2.4 ± 0.3 a	2.2 ± 0.7 a	7.0 ± 0.7 a	4.1 ± 0.3 b	4.1 ± 0.2 b
KW7	2.2 ± 0.0 b	3.6 ± 0.3 a	2.4 ± 0.0 b	7.9 ± 0.0 a	5.0 ± 0.6 b	8.3 ± 0.0 a

Results are presented as the means ± standard deviation (*n* = 3). Different lowercase letters within each result indicate statistically significant differences among applied treatments on each line at *p* ≤ 0.05.

**Table 4 plants-12-02922-t004:** Characterization of carbon nanodots (CND) and manganese ferrite (MnFe_2_O_4_) nanoparticles.

Characteristics	CND	MnFe_2_O_4_ NP
Product name	Carbon nanodots-deep UV fluorescent	Manganese ferrite nanoparticles
Purity	98.5%	98.95%
Average particle size	1.6–1.8 nm	55 nm
Shape of particle	-	Spherical
Physical state	Clear liquid	Powder
Molecular weight	12.011 g/mol	-
Concentration	>250 mg/mL	-
UV-Vis	<190 and 270 nm	-
Emission peak	λ_em._ = 302 and 420 nm by λ_exc._ at 179 nm	-
Photoluminescence quantum yield	11.3%	-
pH value	6.7–7	-

**Table 5 plants-12-02922-t005:** Physiochemical properties of seedbed media.

Seedbed Characteristics	Proportions
Bulk density	0.15–0.25 Mg m^−3^
pH (1:5, *v*/*v*)	5.5–7.0
Electrical conductivity	0.65 ± dS m^−1^
NO_3_^—^N	200–350 mg L^−1^
NH_4_^+^-N	below 150 mg L^−1^
Cation exchange capacity	35–55 cmol^+^ L^−1^
Available phosphorus (P_2_O_5_)	200–350 mg L^−1^
Raw material and mixing ratio (%)	Zeolite 4, perlite 7, vermiculite 6, coco peat 68, peat moss 14.73, fertilizers 0.201, wetting agent 0.064, pH adjusting agent 0.005

## Data Availability

The data is contained within the article.

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
