# Peer review of "Efficacy of Carbon Nanodots and Manganese Ferrite (MnFe_2_O_4_) Nanoparticles in Stimulating Growth and Antioxidant Activity in Drought-Stressed Maize Inbred Lines"

_plants, 2023, doi:10.3390/plants12162922_

Round 1

Reviewer 1 Report

Referee report on the manuscript “Efficacy of carbon nanodots and manganese iron oxide  (MnFe2O4) nanoparticles in stimulating growth and antioxidant  activity in drought-stressed maize inbred lines

The article definitely contains some new and interesting results that can be recommended for publication, but only after the article has been finalized and some shortcomings have been eliminated.

1.  Title.  This part of the title “manganese iron oxide  (MnFe2O4)“ is not correct. It will be better to call this compound as “manganese ferrite” or “MnFe2O4 spinel”. See, for example:

Islam, K.; Haque, M.; Kumar, A.; Hoq, A.; Hyder, F.; Hoque, S.M. Manganese Ferrite Nanoparticles (MnFe2O4): Size Dependence for Hyperthermia and Negative/Positive Contrast Enhancement in MRI. Nanomaterials 202010, 2297.

Brik, M.G.; Ma, C.G.; Yamamoto, T.; Piasecki, M.; Popov, A.I. First-principles methods as a powerful tool for fundamental and applied research in the field of optical materials. In Phosphor Handbook; CRC Press: Boca Raton, FL, USA, 2022; pp. 1–25

2. More information about MnFe2O4, and especially nano- MnFe2O4, would be helpful, especially in light of the comparison and difference in their properties and their new applications. This will attract the attention of more readers. In fact, the choice of the sample is not clear and the motivation to use this particular compound has not been disclosed. 

Author Response

Reviewer 1
The article definitely contains some new and interesting results that can be recommended for publication, but only after the article has been finalized and some shortcomings have been eliminated.

  1. Title.  This part of the title “manganese iron oxide (MnFe2O4)“ is not correct. It will be better to call this compound as “manganese ferrite” or “MnFe2O4 spinel”. See, for example:

Islam, K.; Haque, M.; Kumar, A.; Hoq, A.; Hyder, F.; Hoque, S.M. Manganese Ferrite Nanoparticles (MnFe2O4): Size Dependence for Hyperthermia and Negative/Positive Contrast Enhancement in MRI. Nanomaterials 202010, 2297.

Brik, M.G.; Ma, C.G.; Yamamoto, T.; Piasecki, M.; Popov, A.I. First-principles methods as a powerful tool for fundamental and applied research in the field of optical materials. In Phosphor Handbook; CRC Press: Boca Raton, FL, USA, 2022; pp. 1–25

->We modified the name of manganese iron oxide to manganese ferrite in the title as well as throughout the manuscript.

  1. More information aboutMnFe2O4, and especially nano- MnFe2O4, would be helpful, especially in light of the comparison and difference in their properties and their new applications. This will attract the attention of more readers. In fact, the choice of the sample is not clear and the motivation to use this particular compound has not been disclosed. 

We added a significant amount of information (3rd last paragraph of introduction with blue color) about the properties and applications of MnFe2O4 and compared nano- MnFe2O4 with bulk MnFe2O4. At the start of 2nd last paragraph of the introduction, we comprehend the motive for using this particular compound (red-colored lines).

Reviewer 2 Report

The submitted manuscript to the Plants-MDPI investigates the potential roles of carbon nanodots and manganese iron oxide 2 (MnFe2O4) nanoparticles in improving growth and antioxidant activity in drought-stressed maize inbred lines. Although the topic of manuscript is interesting and under the scope of this journal, however, before further consideration, the following concerns should be resolved:

Line 12-13: Please re-write the sentance.

Line 17-18: According to the sentance structure, please write the names of inbread line in the bracket ().

Line 19: Please always use the round-off values.

The should add 2-3 senatnces about phenolic acids in the abstract section since these are the main components of thier study.

Line 115: Replace „two” with „few”.

Although the graphs are statiscally correct, but I would rather suggest to draw the tables for some of the parameters (e.g. morphological) due to large data-set.

Please explain in detail that how the drought stress was maintained.

Author Response

Reviewer 2
The submitted manuscript to the Plants-MDPI investigates the potential roles of carbon nanodots and manganese iron oxide 2 (MnFe2O4) nanoparticles in improving growth and antioxidant activity in drought-stressed maize inbred lines. Although the topic of the manuscript is interesting and under the scope of this journal, however, before further consideration, the following concerns should be resolved:

Line 12-13: Please re-write the sentence.

->We rewrote that sentence to make it more precise and comprehensive (red color text).

Line 17-18: According to the sentence structure, please write the names of inbread lines in the bracket ().

->We have done it according to the comment (red color text in the Ms).

Line 19: Please always use the round-off values.

->We have rounded off all the values throughout manuscript.

The should add 2-3 senatnces about phenolic acids in the abstract section since these are the main components of thier study.

->We added the sentence about the introduction and function of phenolic acids in plants (line 10-11 red color text)

Line 115: Replace „two” with „few”.

->Replaced word two with “few”.

Although the graphs are statiscally correct, but I would rather suggest to draw the tables for some of the parameters (e.g. morphological) due to large data-set.

->According to the above suggestion, we changed six figures depicting different morphological parameters to 3 tables (Tables 1, 2, and 3).

Please explain in detail how the drought stress was maintained.

->We explained it (blue color text in the “experimental design and crop husbandry” heading).
